# Effects of 12 Weeks of High-, Moderate-, and Low-Volume Training on Performance Parameters in Adolescent Swimmers

Hakan Karabıyık [1], Mehmet Gülü [2,*], Hakan Yapici [2,*], Furkan Iscan [2], Fatma Hilal Yagin [3], Tugay Durmuş [1], Oğuz Gürkan [4], Melek Güler [5], Sinan Ayan [2] and Reem Alwhaibi [6]

[1] Department of Coaching Education, Faculty of Sport Sciences, Ankara University, Ankara 06830, Türkiye; hakankarabiyik7@gmail.com (H.K.); tgydrms@gmail.com (T.D.)
[2] Faculty of Sport Sciences, Kirikkale University, Kırıkkale 71450, Türkiye; furkaniscann@gmail.com (F.I.); sayan@kku.edu.tr (S.A.)
[3] Department of Biostatistics and Medical Informatics, Faculty of Medicine, Inonu University, Malatya 44000, Türkiye; hilal.yagin@inonu.edu.tr
[4] Department of Coaching Education, Faculty of Sport Sciences, Bozok University, Yozgat 66000, Türkiye; oguz.gurkan@bozok.edu.tr
[5] Department of Coaching Education, Faculty of Sport Sciences, Karamanoglu Mehmetbey University, Karaman 70100, Türkiye; melekglr@kmu.edu.tr
[6] Department of Rehabilitation Sciences, College of Health and Rehabilitation Sciences, Princess Nourah bint Abdulrahman University, Riyadh 11671, Saudi Arabia; rmalwhaibi@pnu.edu.sa
* Correspondence: mehmetgulu@kku.edu.tr (M.G.); hakanyapici@kku.edu.tr (H.Y.)

**Abstract:** Swimming, an Olympic sport with diverse distances and energy systems, requires early specialization for elite success. High intensity interval training (HIIT) is a fundamental method used by swimmers to enhance performance, offering both aerobic and anaerobic benefits. This study aimed to examine the effects of a 12-week HIIT program with varying volumes on adolescent swimmers' performance parameters. A total of 50 participants were divided into three groups High Intensity Low Volume (HILV), Moderate Intensity Moderate Volume (MIMV), Low Intensity High Volume (LIHV), and their training sessions consisted of 10 sets with 60 s rest intervals between repetitions and 3 min rest intervals between sets. The intensity was determined based on a pre-test 100 m times. The results indicate significant improvements in anthropometric measurements, including weight, Body Mass Index (BMI), and body fat percentage, within each group, with no significant differences between groups. Swimming performance for various distances (50 m, 100 m, 200 m, and 800 m) showed significant temporal improvements in all groups, with stroke parameters such as stroke rate and length also exhibiting significant improvements ($p < 0.05$). Resting heart rate and swimming performance at 100 m and 200 m differed significantly between groups, highlighting the impact of training volume on specific outcomes. In conclusion, this study highlights the positive impact of interval training on the swimming time (50 m, 100 m, 200 m, and 800 m), stroke rate (SR), strokes per length (SPL), stroke length (SL), Borg scale (BS) for various distances, emphasizing the need for tailored training programs to maximize their development and potential.

**Keywords:** swimming; adolescent athletes; high-intensity interval training

## 1. Introduction

Swimming is a popular Olympic sport that encompasses all energy systems, with events ranging from 50 m to 1500 m distances, divided into 16 separate categories [1]. Swimming races are completed in various durations: short (around 20–30 s), medium (up to 2 min and 30 s), and long (starting from 7 min and extending to hours) [2]. Swimmers are known to train for hours and cover kilometers daily, regardless of their race distances, and it is a well-established fact that many Olympic champions hold world records in various distances. Due to its horizontal position in the water and the unique technique of responding to buoyancy, gravity, and resistance in the water, as well as the different

breathing techniques compared to outdoor sports, swimming is evaluated separately from other sports [3]. Physiologically, it is known that swimming is influenced by lower body fat, broader shoulders and hips, longer stroke length, shorter forearms, and wider arm circumference [4]. To become an elite athlete, early participation in sports is recommended. Swimming includes activities targeting all muscle groups and is one of the sports that needs to be specialized at an early age [5].

Interval training is a fundamental and highly effective method employed by swimmers to boost their performance. This structured training approach involves alternating between periods of high-intensity effort and periods of rest or lower-intensity activity [6]. One of the primary benefits of interval training in swimming is its ability to enhance aerobic and anaerobic capacity [7]. High-intensity interval training is often preferred in swimming to provide aerobic adaptations and enhance performance [8]. Interval training used to improve swimming performance typically involves short duration loads, repetitions close to race pace, and an adequate number of repetitions [9]. It is known that even 4-week interval training in swimmers can enhance parasympathetic modulation and improve autonomic modulation [9]. According to the results of an experimental study, it was found that both short (50 m) and long (100 m) interval exercise protocols over an 8-week period had similar effects on swimming performance, stroke parameters, $VO_{2max}$, and recovery [8].

In light of the significant advantages offered by interval training in enhancing swimmers' performance, it is essential to recognize that swimming is a versatile and inclusive sport suitable for adolescents regardless of their activity level or athletic background [10]. However, adolescents may have varying levels of aerobic and anaerobic capacity due to differences in physical maturity. These variables must be accounted for to understand the differences when assessing training programs and evaluating performance. Also, it should be noted that high-intensity interval exercise leads to different inflammatory markers in adult and adolescent swimmers [11]. Different effects of adult training content on adolescent athletes should also be considered. It is known that in adolescent swimmers, along with swimming exercises, certain changes occur in the conduction system while maintaining systolic and diastolic functions and without changes in left ventricular size [12]. According to another experimental study, unlike adults, the heart rate value in adolescent swimmers has a limited effect on determining the balance between training load and the athlete's tolerance capacity [13]. During the adolescence period, adaptations in athletes become even more crucial when considering the hormonal changes induced by both existing hormonal fluctuations and exercise-induced hormonal changes [14].

Training load is a critical concept in swimming, and it represents the cumulative stress placed on a swimmer's body during training sessions over time [15]. To optimize performance, coaches carefully design training plans that balance the training load with periods of recovery [16]. According to a review, the highest training volume for adolescent swimmers was reported as 17.27 ± 5.25 h/week, whereas for adults, it was 26.8 ± 4.8 h/week. However, in most athletes, it was determined that training at these levels led to shoulder pain [17]. If training loads are not carefully determined in adolescent athletes, it can lead to early burnout, dropping out of school, and the onset of physiological and psychological illnesses [14]. On the other hand, combining swimming with weight-bearing and strength activities during childhood and adolescence promotes better bone development [18].

Adolescents undergo significant physical growth and development, which can impact their swimming performance. Factors such as changes in body composition, muscle development, and bone growth should be considered when assessing and training adolescent swimmers [19]. Applying appropriate training to adolescent swimmers during the adolescence period is crucial for positive effects on performance and overall health [19]. In improving aerobic endurance in adolescent swimmers, it has been reported that interval training with swimming distances at race pace for 200 m and 400 m, along with 30–45 s of rest intervals, can be effective [20]. A 15-week interval training program, whether with a low or high volume, has been found to increase stroke length, stroke rate, and stroke count in adolescent swimmers [21]. Considering the similar physiological effects of 100 m

and 200 m interval training at the maximum aerobic speed in adolescent swimmers [22], it is worth considering that varying the volume of training could alter the resulting physiological effects. Wahyudi [23] suggests that aerobic interval training positively impacts the performance of early adolescent freestyle swimmers. In another study, Kabasakalis et al. [24] suggest that sprint interval swimming exercise had a modest but potentially beneficial impact on irisin levels and redox status markers in adolescent swimmers. Pinos et al. [25] demonstrate that both pool-based sprint training and dry land (ergometer) training led to significant improvements in anaerobic sprint ability among adolescent swimmers. This suggests that using multiple training modalities (both in the pool and on dry land) can be effective in enhancing sprint performance in swimmers.

Swimming, as a sport that spans various age categories such as children and adolescents, could be an ideal field for investigating the effects of high-intensity interval training [26]. Despite being a popular sport, the challenges in obtaining physiological measurements and the individual effects of water on athletes contribute to the presence of fewer studies in the literature compared to other sports like athletics and cycling [27]. Additionally, as far as we know, interval training at low, moderate, and high volumes has not been extensively studied in adolescent swimmers. In the context of this research investigation, our hypothesis centered around the notion that employing a training regimen characterized by high-intensity workouts paired with a reduced volume of training sessions would yield more favorable outcomes among adolescent swimmers. In the context of this research, our hypothesis centered around the notion that employing a training regimen characterized by high-intensity workouts paired with a reduced volume of training sessions would yield more favorable outcomes among adolescent swimmers. We believed that this unique approach to training had the potential to offer significant advantages and improvements when compared to conventional training methods commonly employed in the sport of swimming for this age group. By emphasizing high-intensity exercises while reducing the overall training load, we aimed to uncover the potential benefits and optimize performance for young athletes. This study aims to investigate the effects of a 12-week interval training program with varying volumes on the performance parameters in adolescent swimmers.

## 2. Materials and Methods

In this study, the effect of swimming training applied at three different volumes on performance parameters was investigated. This study was carried out at Kırıkkale University Faculty of Sport Sciences according to the principles stated in the Declaration of Helsinki [28]. After it was approved by Kırıkkale University Non-Interventional Research Ethics Committee (Date: 12 January 2022, number: 2022.01.04), parents and coaches were fully informed about the protocols of this study and the purpose of this study, and the athletes were allowed to participate voluntarily by signing a written consent form. The equivalence of the athletes participating in this study in free swimming style and track exit skills was tested, and students with the same swimming level were selected for this study.

### 2.1. Participants

Working group: 50 adolescents between the ages of 13–14 who were licensed to swim in the Private Sports Club in Kırıkkale (Turkey) [mean age: $13.54 \pm 0.50$ years, average body fat ratio: $21.71 \pm 4.47$, average body weight: $54.92 \pm 10.41$ kg, average height: $158.80 \pm 11.29$ cm; Body Mass Index (BMI) values are $21.74 \pm 3.30$ kg/m$^2$]. Before participating in our study, the participants were completing an average of 4000 m of swimming training per week. The HILV group performed 4 repetitions of 50 m swimming at an intensity range of 85–95% on two days of the week (with 60 s of rest between repetitions) for a total of 10 sets (with 3 min of rest between sets). The MIMV group conducted 3 repetitions of 50 m swimming at an intensity range of 75–85% on three days of the week (with 60 s of rest between repetitions) for a total of 10 sets (with 3 min of rest between sets). The LIHV group executed 2 repetitions of 50 m swimming at an intensity

range of 65–75% on five days of the week (with 60 s of rest between repetitions) for a total of 10 sets (with 3 min of rest between sets). The training volumes for the HILV, MIMV, and LIHV groups are 4000 m, 4500 m, and 5000 m, respectively. Each group also performed a standard 400 m warm-up and a 200 m cooldown. In the participants: Anthropometric tests (height, body weight, BMI), 50 m, 100 m, 200 m and 800 m swimming performance specific to the sport branch, stroke rate (SR), stroke per length (SPL), stroke length (SL), $VO_{2max}$, and strength measurements were made. The pre-test and post-test measurements of the swimmers who participated in the measurements were taken by the same researchers, and none of them were excluded from this study.

### 2.2. Design and Procedures

The entire group of swimmers studied participated in the same 12-week training under the supervision of the same coaches at the same training facility. Test measurements were made in Kırıkkale Olympic Swimming Pool. Identical conditions prevailed during each experimental series, which took place at the same time of day. Pool ambient temperature: 25–29 °C, pool water temperature: 24–28 °C, pool water chlorine (free): 1–1.5 ppm, humidity absolute: 14 g/kg, humidity relative: ~40–50%, and pool water PH: 7.2–7.6. In research measurements: A portable height meter (SECA 217, Germany) with a precision of 0.01 m for height and an electronic scale with a precision of 0.1 kg were used for body weight measurement. Standard procedures were followed for each assessment test. Anthropometric measurements followed by physical performance tests were collected by the same trained team. All data of the participants were made by a single observer according to the anthropometric measurements reference guide. Before starting the tests, the athletes were asked to complete a 20 min warm-up procedure that included a 10-min warm-up and a 10-min stretch. Participants were instructed to complete each training process with maximum effort. All participants were tested in a specific order to standardize the testing process. Physical fitness, sports skills (anthropometric tests, 50 m, 100 m, 200 m and 800 m swimming performance, stroke rate (SR), stroke per length (SPL), stroke length (SL), heart rate, strength measurement, free swimming styles), and indoor swimming were evaluated in the pool.

### 2.3. Measurements

2.3.1. Anthropometric Measurements

Height, body mass, and body composition measurements were performed using the Bioelectrical Impedance Analysis (BIA) (Tanita Body Composition Analyzer BC 418 Professional model, Tokyo, Japan). The subjects' heights were measured with an anthropometrics rod set on the test day before performance tests to the nearest 0.1 kg and 1 cm. Anthropometric measurements measured standing height and body weight with light clothing and no shoes. We preferred using BIA testing to estimate body fat percentages, guided by research studies that have demonstrated its validity. In this direction, BMI values were calculated by measuring the height and weight of the individuals.

2.3.2. Measuring Swimming Time (50, 100, 200, and 800 m)

Measurements were taken in a 50 m full Olympic swimming pool. When the athlete felt ready, the chronometer started with the exit from the sprint stone and stopped at the end of the distance. Athletes were encouraged to swim the distance they swam at maximum capacity. Then 50 m, 100 m, and 200 m degrees were taken three times, and the best one of these degrees was recorded in the measurement table. After each measurement, a 15 min break was taken for recovery. The 800 m degree was taken and recorded only once.

2.3.3. Stroke Rate (SR)

It is known that stroke rate is an important variable in reaching maximal performance [29]. SR expresses the number of strokes taken in 1 min [30]. Based on any arm of the athlete, time was started from the moment he entered the water with the help of a

stopwatch. The time elapsed when the same arm came out of the water for the 3rd time and was stopped (for example, stroke rate per minute (SR) = 60/1.067 = 57 in the athlete who beat 3 arms in 3.2 s: 3.2 s/3 arms = 1.067 s/arm). After the athlete completed the distance under the water and started the first stroke, the stopwatch was started, and the stopwatch was stopped when the third stroke came out of the water, and it was calculated according to the formula.

### 2.3.4. Strokes Per Length (SPL)

It is the total number of strokes made at a certain distance (for example, an athlete who takes 28 strokes at a distance of 50 m has an SPL value of 28). In the scope of the research: The distance was determined as 50 m, and the athlete started swimming with the whistle command, with one arm holding the sprint stone behind and the other arm reaching forward in the pool. The number of strokes made during the distance was counted and recorded.

### 2.3.5. Stroke Length (SL)

SL, which is seen as one of the most important parameters in measuring success in swimming [31,32], is defined as the stroke distance, and the calculated figure is personal. SL calculation is made by dividing the total distance by the number of strokes taken at that distance (Distance (Distance)/SPL). Although there are 2 variables in the formula, there are many hidden variables, such as your fathom length (the distance between the fingertips of your two hands when you open your arms), shoulder width, hand and foot size, gender, and age. In particular, the issue of stroke length is seen as a determining factor. While measuring SL within the scope of this research, distance and SPL degrees were calculated according to the formula.

### 2.3.6. Borg Scale (BS) 50 m, 100 m, 200 m and 800 m

It contains a value between 6 and 20 (corresponding to the normal heart rate range of 60 to 200) and used to measure overall effort during physical activity [33].

### 2.3.7. Resting Heart Rate

Heart rate is defined as the count of heartbeats occurring within a single minute. Resting heart rate measurements were taken while participants were in sitting position, at complete rest, in a relaxed state, and not engaged in any physical or mental activities that could increase their heart rate. We measured resting heart rates by using the Polar H10 heart rate sensors (Polar Electro, Kempele, Finland).

### 2.3.8. Trainings

The total duration of the training program was 12 weeks. Participants were allocated into three groups using a counterbalanced approach considering their pre-test results. In all groups, each training session comprised 10 sets of freestyle swimming. There was a 60 s rest period between repetitions and a 3 min rest period between sets. The training intensity levels were determined based on the 100 m times recorded during the pre-test. HILV group: This group engaged in four repetitions of 50 m swimming at an intensity range of 85–95% on two days per week with a 60 s rest between repetitions. Each session consisted of a total of 10 sets with a 3 min rest between sets. The total training volume for the HILV group was 4000 m. MIMV group: The MIMV group performed three repetitions of 50 m swimming at an intensity range of 75–85% on three days per week with a 60 s rest between repetitions. Similar to the HILV group, each session included 10 sets with a 3 min rest between sets. The total training volume for the MIMV group was 4500 m. LIHV group: The LIHV group executed two repetitions of 50 m swimming at an intensity range of 65–75% on five days per week with a 60 s rest between repetitions. Each session comprised 10 sets with a 3 min rest between sets. The total training volume for the LIHV group was 5000 m. Additionally, each group incorporated a standardized 400 m warm-up and a 200 m cooldown into their

training routine. At the end of each repetition, participants' perceived exertion levels and heart rate values were recorded. Training protocol is given in Table 1.

**Table 1.** Training protocol.

| Group | HILV | MIMV | LIHV |
|---|---|---|---|
| Warm-up | 400 m | 400 m | 400 m |
| Sets (Repetitions) | $10 \times (4 \times 50 \text{ m})$ | $10 \times (3 \times 50 \text{ m})$ | $10 \times (2 \times 50 \text{ m})$ |
| Recovery | 200 m | 200 m | 200 m |
| Rest Between Intervals | 60 s | 60 s | 60 s |
| Rest Between Sets | 3 min | 3 min | 3 min |
| Intensity | 85–95% | 75–85% | 65–75% |
| Frequency (Days) | 2 | 3 | 5 |

Load has set as % based on the best 100 m score. Only freestyle swimming was used in the training. HILV = High Intensity Low Volume; MIMV = Moderate Intensity Moderate Volume; LIHV = Low Intensity High Volume.

### 2.4. Power Analysis

G*power software was used to determine the appropriate sample size for this study. As a result (alpha value = 0.05 and 1-beta value = 0.80, $\eta_p^2 = 0.25$), it was calculated that at least twelve swimmers should be included in this study [34].

### 2.5. Data Analysis

In this study, the assumption of normal distribution of quantitative variables was examined via visual (histogram and probability graphs) and analytical (Shapiro–Wilk Test) methods. Quantitative variables were expressed as mean and standard deviation since they showed a normal distribution [35]. In order to examine the results of different protocols (HILV, MIMV, and LIHV), pre- and post-test measurements, and protocol*time interaction effect, repeated measures were determined using two-way ANOVA test. Mauchly sphericity test was used to test the homogeneity of variances, and Greenhouse–Geisser correction was applied when necessary. Partial eta-squares ($\eta_p^2$) were calculated for the magnitude of the effect between groups. When statistically significant differences were discovered between study protocols, multiple comparison analyses were performed using the Tukey method. After post-hoc analyses, Cohen's d was used for effect size (ES) calculations. The magnitude of effect size was considered following the thresholds: Cohen suggested that d = 0.2 be considered a "small" effect size, 0.5 represents a "medium" effect size, and 0.8 a "large" effect size [36], $p < 0.05$ was considered significant. American Psychological Association (APA) 6.0 style was used to report statistical differences [37]. All analyses were performed using IBM SPSS Statistics for Windows version 28.0 (New York, NY, USA).

## 3. Results

The results regarding the anthropometric data of the participants are given in Table 2. When Table 2 was examined, there was a main effect of time on the weight, BMI, and fat results ($p < 0.001$). According to the effect size results for the time factor, it was seen that the smallest effect was in the fat (%) value ($\eta_p^2 = 0.42$), and the largest effect was in the BMI values ($\eta_p^2 = 0.89$). The protocols did not affect the weight, BMI, and fat results ($p > 0.05$). Furthermore, the protocol*time interaction effect was significant for the fat outcomes ($p < 0.001$), and the fat outcomes were lowest in the post-test with the MIMV protocol but not significant for the weight and BMI outcomes ($p > 0.05$) (Table 2).

Table 3 shows the participants' 50 m, 100 m, 200 m, and 800 m swimming results. According to Table 3, the time main effect was significant for the 50 m, 100 m, and 800 m swimming results ($p < 0.001$), and swimming times improved in the post-test. However, the main effect of time and the interaction effect of protocol*time were not significant for 200 m swimming results ($p > 0.05$). The effect of the protocols was significant for the 50 m ($\eta_p^2 = 0.86$), 100 m ($\eta_p^2 = 0.79$), 200 m ($\eta_p^2 = 0.60$), and 800 m ($\eta_p^2 = 0.62$) swimming results, and the results improved with the HILV protocol. The protocol*time interaction effect was

statistically significant ($p < 0.001$) for the 50 m, 100 m, and 800 m results, and the results showed improved swimming results post-test with the HILV protocol.

**Table 2.** Evaluation of change and percentage values of body composition measurements before and after swim training.

| $n = 50$ | Pre | Post | Δ | % | | Two-Way Repeated ANOVA | | | |
|---|---|---|---|---|---|---|---|---|---|
| Variable | M ± SD | M ± SD | $T_B T_{end}$ | $T_B - T_{end}$ | $\eta_p^2$ | Protocol*Time | Time | Protocol | Pairwise Comparisons (*p*); (ES) |
| Weight (Kg) | | | | | | | | | |
| HILV(16) | 59.6 ± 2.5 | 58.7 ± 2.5 | −0.9 ± 0.4 | 1.7 | 0.704 | F = 1.9 | F = 387.8 | F = 2.6 | |
| MIMV(16) | 53.3 ± 2.5 | 52.3 ± 2.5 | −1.1 ± 0.3 | 1.8 | 0.774 | $p = 0.163$ | $p < 0.001$ * | $p = 0.08$ * | - |
| LIHV(18) | 52.2 ± 2.4 | 51.4 ± 2.4 | −0.8 ± 0.3 | 1.5 | 0.714 | $\eta_p^2 = 0.07$ | $\eta_p^2 = 0.88$ | $\eta_p^2 = 0.09$ | |
| BMI (Kg/m²) | | | | | | | | | |
| HILV(16) | 22.7 ± 3.0 | 22.3 ± 2.9 | 0.33 ± 0.1 | 1.8 | 0.696 | F = 1.1 | F = 390.7 | F = 1.5 | |
| MIMV(16) | 20.7 ± 2.9 | 20.3 ± 2.9 | 0.40 ± 0.1 | 2.0 | 0.765 | $p = 0.339$ | $p < 0.001$ * | $p = 0.232$ | - |
| LIHV(18) | 21.8 ± 3.8 | 21.5 ± 3.8 | 0.35 ± 0.2 | 1.4 | 0.738 | $\eta_p^2 = 0.04$ | $\eta_p^2 = 0.89$ | $\eta_p^2 = 0.06$ | |
| Fat (%) | | | | | | | | | |
| HILV(16) | 22.9 ± 4.9 | 19.2 ± 3.5 | 3.75 ± 2.0 | 19.2 | 0.746 | F = 16.7 | F = 157.7 | F = 0.35 | |
| MIMV(16) | 20.7 ± 4.3 | 19.0 ± 4.0 | 1.71 ± 0.7 | 8.9 | 0.378 | $p < 0.001$ * | $p < 0.001$ * | $p = 0.707$ | - |
| LIHV(18) | 21.5 ± 4.9 | 20.2 ± 4.5 | 1.37 ± 0.7 | 6.4 | 0.304 | $\eta_p^2 = 0.42$ | $\eta_p^2 = 0.77$ | $\eta_p^2 = 0.01$ | |

M: Mean; SD: Standard deviation; HILV = High Intensity Low Volume; MIMV = Moderate Intensity Moderate Volume; LIHV = Low Intensity High Volume; Δ = Change; Pre = Preintervention; Post = Postintervention; $\eta_p^2$: Partial eta squared; * denotes statistical significance at the $p < 0.05$ level; ES: Effect size.

When the results of stroke, SR, SPL, and SL were examined, the interaction effect of protocols, time, and protocol*time was significant in all stroke results ($p < 0.05$) (Table 4). The interaction results in stroke, SR, and SL showed the best improvement in the pre-test with the LIHV protocol, but the SPL results showed better improvement in the post-test with the HILV protocol. The effect size results for the interaction were found to have the greatest effect on the stroke results ($\eta_p^2 = 0.58$), and the smallest effect on the SPL results ($\eta_p^2 = 0.19$). When the effect size results in the time factor were examined, it was determined that the weakest effect was on the SL results ($\eta_p^2 = 0.66$), and the largest effect was on the stroke results ($\eta_p^2 = 0.90$). While the weakest effect in the protocols was seen in the stroke results ($\eta_p^2 = 0.41$), the greatest effect was seen in the SL results ($\eta_p^2 = 0.81$) (Table 4).

When the resting HR, HR50 m, HR100 m, HR200 m, and HR800 m results were examined, the main effect of time was significant ($p < 0.001$), and heart rate results were lower in the post-test compared to the pre-test. The greatest effect on the time factor was observed in the HR50 m results ($\eta_p^2 = 0.89$). The effect of the protocols was statistically significant for HR100 m and HR200 m ($p < 0.05$) but not for HR, HR50 m, and HR800 m ($p > 0.05$). The protocol*time interaction effect was significant for HR50 m and HR100 m, and the results were lowest in the post-test with the LIHV protocol ($p < 0.05$). However, no interaction effect was detected at resting HR, HR200 m, HR800 m ($p > 0.05$) (Table 5).

The main effects of time and the protocols were statistically significant on all Borg scale results ($p < 0.05$). Moreover, the BS100 m, BS200 m, and BS800 m results had a protocol*time interaction effect, and the results improved post-test results with the HILV protocol. However, the interaction effect was not significant for BS50 m ($p > 0.05$). When the effect size results in the time factor were examined, it was determined that the weakest effect was on the BS200 m results ($\eta_p^2 = 0.51$), and the highest effect was on the BS100 m swimming results ($\eta_p^2 = 0.76$). In the protocols, the smallest effect was seen in the BS200 m results ($\eta_p^2 = 0.32$), while the greatest effect was seen in the BS800 m results ($\eta_p^2 = 0.75$) (Table 6).

**Table 3.** Evaluation of change and percentage values with 50 m, 100 m, 200 m and 800 m swimming measurements before and after swimming training.

| *n* = 50 | Pre | Post | Δ | % | $\eta_p^2$ | Two-Way Repeated ANOVA | | | |
|---|---|---|---|---|---|---|---|---|---|
| Variable | M ± SD | M ± SD | $T_B - T_{end}$ | $T_B - T_{end}$ | | Protocol*Time | Time | Protocol | Pairwise Comparisons (*p*); (ES) |
| 50 m swimming (s) | | | | | | | | | |
| HILV(16) | 33.8 ± 1.8 | 31.4 ± 1.6 | 2.4 ± 0.6 | 7.6 | 0.814 | F = 4.8 | F = 450.7 | F = 148.9 | HILV-MIMV: <0.001; 1.05 |
| MIMV(16) | 38.2 ± 1.0 | 36.3 ± 1.1 | 1.9 ± 0.3 | 5.2 | 0.743 | *p* = 0.012 | *p* < 0.001 * | *p* < 0.001 * | HILV-LIHV: <0.001; 0.91 |
| LIHV(18) | 40.8 ± 1.2 | 39.1 ± 0.9 | 1.7 ± 0.9 | 4.3 | 0.709 | $\eta_p^2$ = 0.17 | $\eta_p^2$ = 0.91 | $\eta_p^2$ = 0.86 | MIMV-LIHV:<0.001; 0.30 |
| 100 m swimming (min) | | | | | | | | | |
| HILV(16) | 1.12 ± 0.04 | 1.08 ± 0.04 | 0.04 ± 0.01 | 3.7 | 0.838 | F = 23.7 | F = 375.5 | F = 93.2 | HILV-MIMV: <0.001; 1.00 |
| MIMV(16) | 1.20 ± 0.03 | 1.17 ± 0.02 | 0.03 ± 0.01 | 1.7 | 0.726 | *p* < 0.001 * | *p* < 0.001 * | *p* < 0.001 * | HILV-LIHV: <0.001; 2.00 |
| LIHV(18) | 1.25 ± 0.03 | 1.23 ± 0.03 | 0.02 ± 0.01 | 1.6 | 0.476 | $\eta_p^2$ = 0.50 | $\eta_p^2$ = 0.88 | $\eta_p^2$ = 0.79 | MIMV-LIHV:<0.001; 1.00 |
| 200 m swimming (min) | | | | | | | | | |
| HILV(16) | 2.28 ± 0.8 | 2.22 ± 0.8 | 0.06 ± 0.1 | 2.7 | 0.050 | F = 1.4 | F = 1.3 | F = 35.7 | HILV-MIMV: 0.005; 3.00 |
| MIMV(16) | 2.46 ± 0.6 | 2.43 ± 0.7 | 0.03 ± 0.1 | 1.2 | 0.019 | *p* = 0.274 | *p* = 0.259 | *p* < 0.001 * | HILV-LIHV: <0.001; 4.00 |
| LIHV(18) | 2.73 ± 0.2 | 2.71 ± 0.3 | 0.02 ± 0.1 | 0.7 | 0.008 | $\eta_p^2$ = 0.05 | $\eta_p^2$ = 0.03 | $\eta_p^2$ = 0.60 | MIMV-LIHV:<0.001; 1.00 |
| 800 m swimming (min) | | | | | | | | | |
| HILV(16) | 10.8 ± 0.4 | 10.4 ± 0.4 | 0.32 ± 0.4 | 3.8 | 0.350 | F = 3.3 | F = 29.7 | F = 39.1 | HILV-MIMV: 0.005; 5.14 |
| MIMV(16) | 11.3 ± 0.1 | 11.2 ± 0.2 | 0.17 ± 0.1 | 1.0 | 0.134 | *p* = 0.046 | *p* < 0.001 * | *p* < 0.001 * | HILV-LIHV: <0.001; 7.88 |
| LIHV(18) | 12.3 ± 0.8 | 12.2 ± 0.8 | 0.09 ± 0.1 | 0.8 | 0.054 | $\eta_p^2$ = 0.12 | $\eta_p^2$ = 0.38 | $\eta_p^2$ = 0.62 | MIMV-LIHV:<0.001; 8.00 |

M: Mean; SD: Standard deviation; HILV = High Intensity Low Volume; MIMV = Moderate Intensity Moderate Volume; LIHV = Low Intensity High Volume; Δ = Change; Pre = Preintervention; Post = Postintervention; $\eta_p^2$: Partial eta squared; * denotes statistical significance at the *p* < 0.05 level; NS: not significant; ES: Effect size.

**Table 4.** Evaluation of change and percentage values with SR, SPL, and SL swim measurements before and after swim training.

| *n* = 50 | Pre | Post | Δ | % | $\eta_p^2$ | Two-Way Repeated ANOVA | | | |
|---|---|---|---|---|---|---|---|---|---|
| Variable | M ± SD | M ± SD | $T_B - T_{end}$ | $T_B - T_{end}$ | | Protocol*Time | Time | Protocol | Pairwise Comparisons (*p*); (ES) |
| Stroke | | | | | | | | | |
| HILV(16) | 70.2 ± 5.7 | 75.9 ± 5.2 | 5.7 ± 1.5 | 8.1 | 0.871 | F = 32.5 | F = 416.4 | F = 16.4 | HILV vs. MIMV: 0.001; 2.08 |
| MIMV(16) | 64.6 ± 5.1 | 67.8 ± 5.2 | 3.2 ± 0.8 | 4.9 | 0.679 | *p* < 0.001 * | *p* < 0.001 * | *p* < 0.001 * | HILV vs. LIHV: <0.001; 2.41 |
| LIHV(18) | 61.8 ± 5.5 | 64.0 ± 4.8 | 2.2 ± 1.4 | 3.6 | 0.536 | $\eta_p^2$ = 0.58 | $\eta_p^2$ = 0.90 | $\eta_p^2$ = 0.41 | MIMV vs. LIHV:0.174; NS |

**Table 4.** *Cont.*

| n = 50 | Pre | Post | Δ | % | $\eta_p^2$ | Two-Way Repeated ANOVA | | | |
|---|---|---|---|---|---|---|---|---|---|
| Variable | M ± SD | M ± SD | $T_B - T_{end}$ | $T_B - T_{end}$ | | Protocol*Time | Time | Protocol | Pairwise Comparisons (p); (ES) |
| | | | | Stroke Rate | | | | | |
| HILV(16) | 56.1 ± 4.2 | 59.9 ± 3.8 | 3.8 ± 1.7 | 6.8 | 0.771 | F = 10.6 | F = 257.9 | F = 17.9 | HILV vs. MIMV: 0.421; NS |
| MIMV(16) | 55.3 ± 3.3 | 57.9 ± 3.1 | 2.6 ± 1.0 | 4.7 | 0.600 | p < 0.001 * | p < 0.001* | p < 0.001 * | HILV vs. LIHV: <0.001; 1.39 |
| LIHV(18) | 50.8 ± 2.6 | 52.7 ± 2.4 | 1.9 ± 0.9 | 3.7 | 0.489 | $\eta_p^2 = 0.31$ | $\eta_p^2 = 0.85$ | $\eta_p^2 = 0.43$ | MIMV vs. LIHV: <0.001; 0.73 |
| | | | | Strokes Per Length | | | | | |
| HILV(16) | 42.44 ± 2.7 | 40.88 ± 2.4 | −1.6 ± 0.9 | 3.7 | 0.589 | F = 5.7 | F = 95.2 | F = 100.4 | HILV vs. MIMV: <0.001; 0.87 |
| MIMV(16) | 47.88 ± 2.8 | 47.00 ± 2.4 | −0.9 ± 0.7 | 1.9 | 0.310 | p = 0.006 * | p < 0.001 * | p < 0.001 * | HILV vs. LIHV:<0.001; 1.12 |
| LIHV(18) | 56.22 ± 3.7 | 55.50 ± 3.4 | −0.7 ± 0.7 | 1.3 | 0.256 | $\eta_p^2 = 0.19$ | $\eta_p^2 = 0.67$ | $\eta_p^2 = 0.80$ | MIMV vs. LIHV: <0.001; 0.28 |
| | | | | Stroke Length | | | | | |
| HILV(16) | 1.18 ± 0.07 | 1.23 ± 0.07 | −0.04 ± 0.2 | 4.2 | 0.696 | F = 17.2 | F = 103.4 | F = 103.4 | HILV vs. MIMV: <0.001; 1.0 |
| MIMV(16) | 1.05 ± 0.06 | 1.07 ± 0.05 | −0.02 ± 0.2 | 1.9 | 0.285 | p < 0.001 * | p < 0.001 * | p < 0.001 * | HILV vs. LIHV: <0.001; 1.89 |
| LIHV(18) | 0.89 ± 0.06 | 0.90 ± 0.06 | −0.01 ± 0.1 | 1.1 | 0.139 | $\eta_p^2 = 0.42$ | $\eta_p^2 = 0.66$ | $\eta_p^2 = 0.81$ | MIMV vs. LIHV: <0.001; 0.63 |

M: Mean; SD: Standard deviation; LIHV = Low volume swimming training; MIMV = Moderate volume swimming training; HILV = High volume swimming training; Δ = Change; Pre = Preintervention; Post = Postintervention; $\eta_p^2$: Partial eta squared; * denotes statistical significance at the $p < 0.05$ level; NS: not significant; ES: Effect size.

**Table 5.** Evaluation of change and percentage values with pulse swim measurements before and after swim training.

| n = 50 | Pre | Post | Δ | % | $\eta_p^2$ | Two-Way Repeated ANOVA | | | |
|---|---|---|---|---|---|---|---|---|---|
| Variable | M ± SD | M ± SD | $T_B - T_{end}$ | $T_B - T_{end}$ | | Protocol*Time | Time | Protocol | Pairwise Comparisons (p); (ES) |
| | | | | Resting HR | | | | | |
| HILV(16) | 93 ± 5.164 | 85.5 ± 4.351 | 7.5 ± 3.2 | 8.8 | 0.662 | F = 0.5 | F = 272.4 | F = 0.35 | |
| MIMV(16) | 91.8 ± 7.1 | 84 ± 5.8 | 7.7 ± 2.7 | 9.3 | 0.677 | p = 0.571 | p < 0.001 * | p = 0.701 | - |
| LIHV(18) | 91.3 ± 5.4 | 84.7 ± 3.9 | 6.7 ± 3.3 | 7.8 | 0.635 | $\eta_p^2 = 0.02$ | $\eta_p^2 = 0.85$ | $\eta_p^2 = 0.01$ | |
| | | | | HR50 m | | | | | |
| HILV(16) | 153.8 ± 8.4 | 141.8 ± 7.3 | 12.0 ± 3.3 | 8.5 | 0.858 | F = 22.7 | F = 409.8 | F = 2.6 | |
| MIMV(16) | 148 ± 8.4 | 141.3 ± 8.6 | 6.7 ± 2.8 | 4.7 | 0.656 | p < 0.001 * | p < 0.001 * | p = 0.08 | - |
| LIHV(18) | 144.4 ± 7.9 | 138.7 ± 7.4 | 5.8 ± 2.5 | 4.1 | 0.611 | $\eta_p^2 = 0.49$ | $\eta_p^2 = 0.89$ | $\eta_p^2 = 0.10$ | |

**Table 5.** *Cont.*

| n = 50 | Pre | Post | Δ | % | $\eta_p^2$ | Two-Way Repeated ANOVA | | | |
|---|---|---|---|---|---|---|---|---|---|
| Variable | M ± SD | M ± SD | $T_B - T_{end}$ | $T_B - T_{end}$ | | Protocol*Time | Time | Protocol | Pairwise Comparisons (*p*); (ES) |
| | | | | | HR100 m | | | | |
| HILV(16) | 147.3 ± 10.2 | 137.5 ± 9.6 | 9.7 ± 2.9 | 7.1 | 0.352 | F = 9.3 | F = 285.0 | F = 4.7 | HILV vs. MIMV: 0.260; NS |
| MIMV(16) | 141 ± 8.6 | 132.8 ± 8.3 | 8.2 ± 4.0 | 6.2 | 0.001 | *p* < 0.001 * | *p* < 0.001 * | *p* = 0.013 | HILV vs. LIHV: 0.011; 1.64 |
| LIHV(18) | 135.6 ± 8.8 | 130.4 ± 8.7 | 5.1 ± 2.7 | 4.0 | 0.163 | $\eta_p^2 = 0.28$ | $\eta_p^2 = 0.86$ | $\eta_p^2 = 0.17$ | MIMV vs. LIHV: 0.633; NS |
| | | | | | HR200 m | | | | |
| HILV(16) | 131.8 ± 9.3 | 126 ± 9.2 | 5.8 ± 5.4 | 4.6 | 0.428 | F = 2.0 | F = 137.1 | F = 7.1 | HILV vs. MIMV: 0.019; 0.49 |
| MIMV(16) | 125.5 ± 6.7 | 117.5 ± 6.2 | 8.0 ± 3.3 | 6.8 | 0.591 | *p* = 0.145 | *p* < 0.001 * | *p* = 0.002 * | HILV vs. LIHV: 0.002; 0.05 |
| LIHV(18) | 122.7 ± 6.7 | 117.1 ± 6.8 | 5.6 ± 2.4 | 4.8 | 0.440 | $\eta_p^2 = 0.08$ | $\eta_p^2 = 0.74$ | $\eta_p^2 = 0.23$ | MIMV vs. LIHV: 1; NS |
| | | | | | HR800 m | | | | |
| HILV(16) | 112.3 ± 5.7 | 105.8 ± 4.6 | 6.5 ± 2.5 | 6.1 | 0.631 | F = 1.5 | F = 233.1 | F = 0.2 | |
| MIMV(16) | 111.3 ± 5.9 | 104.3 ± 5.4 | 7.0 ± 2.7 | 6.7 | 0.664 | *p* = 0.236 | *p* < 0.001 * | *p* = 0.82 | - |
| LIHV(18) | 110.9 ± 6.7 | 105.6 ± 6.3 | 5.3 ± 3.4 | 5.0 | 0.564 | $\eta_p^2 = 0.06$ | $\eta_p^2 = 0.83$ | $\eta_p^2 = 0.008$ | |

M: Mean; SD: Standard deviation; HR: Heart rate; HILV = High Intensity Low Volume; MIMV = Moderate Intensity Moderate Volume; LIHV = Low Intensity High Volume; Δ = Change; Pre = Preintervention; Post = Postintervention; $\eta_p^2$: Partial eta squared; * denotes statistical significance at the *p* < 0.05 level; NS: not significant.

**Table 6.** Evaluation of change and percentage values with Borg scale measurements before and after swim training.

| n = 50 | Pre | Post | Δ | % | $\eta_p^2$ | Two-Way Repeated ANOVA | | | |
|---|---|---|---|---|---|---|---|---|---|
| Variable | M ± SD | M ± SD | $T_B - T_{end}$ | $T_B - T_{end}$ | | Protocol*Time | Time | Protocol | Pairwise Comparisons (*p*); (ES) |
| | | | | | BS 50 m | | | | |
| HILV(16) | 3.38 ± 0.5 | 2.56 ± 0.5 | 0.81 ± 0.4 | 32.0 | 0.498 | F = 2.5 | F = 86.7 | F = 35.6 | HILV vs. MIMV: 0.153; NS |
| MIMV(16) | 3.62 ± 0.6 | 3.00 ± 0.6 | 0.63 ± 0.5 | 20.7 | 0.370 | *p* = 0.09 | *p* < 0.001 * | *p* < 0.001 * | HILV vs. LIHV: <0.001; 8.17 |
| LIHV(18) | 4.61 ± 0.5 | 4.17 ± 0.6 | 0.44 ± 0.5 | 10.6 | 0.251 | $\eta_p^2 = 0.10$ | $\eta_p^2 = 0.65$ | $\eta_p^2 = 0.60$ | MIMV vs. LIHV: <0.001; 3.80 |
| | | | | | BS 100 m | | | | |
| HILV(16) | 4.25 ± 0.6 | 3.13 ± 0.5 | 1.13 ± 0.5 | 35.8 | 0.638 | F = 14.7 | F = 151.1 | F = 20.3 | HILV vs. MIMV: 0.555; NS |
| MIMV(16) | 4.50 ± 0.6 | 3.38 ± 0.5 | 1.13 ± 0.5 | 33.1 | 0.638 | *p* < 0.001 * | *p* < 0.001 * | *p* < 0.001 * | HILV vs. LIHV: <0.001; 16.0 |
| LIHV(18) | 4.94 ± 0.6 | 4.61 ± 0.6 | 0.33 ± 0.5 | 7.2 | 0.148 | $\eta_p^2 = 0.39$ | $\eta_p^2 = 0.76$ | $\eta_p^2 = 0.46$ | MIMV vs. LIHV: <0.001; 16.0 |

**Table 6.** *Cont.*

| *n* = 50 | Pre | Post | Δ | % | $\eta_p^2$ | Two-Way Repeated ANOVA | | | |
|---|---|---|---|---|---|---|---|---|---|
| Variable | M ± SD | M ± SD | $T_B - T_{end}$ | $T_B - T_{end}$ | | Protocol*Time | Time | Protocol | Pairwise Comparisons (*p*); (ES) |
| BS 200 m | | | | | | | | | |
| HILV(16) | 4.88 ± 0.7 | 4.13 ± 0.7 | 0.75 ± 0.5 | 18.1 | 0.178 | F = 4.7 | F = 49.4 | F = 11.3 | HILV vs. MIMV: 0.740; NS |
| MIMV(16) | 4.44 ± 0.6 | 4.06 ± 0.7 | 0.38 ± 0.5 | 9.4 | 0.465 | *p* = 0.01 * | *p* < 0.001 * | *p* < 0.001 * | HILV vs. LIHV:0.005; 9.4 |
| LIHV(18) | 5.33 ± 0.6 | 5.06 ± 0.5 | 0.28 ± 0.5 | 5.3 | 0.118 | $\eta_p^2$ = 0.16 | $\eta_p^2$ = 0.51 | $\eta_p^2$ = 0.32 | MIMV vs. LIHV: <0.001; 2.0 |
| BS 800 m | | | | | | | | | |
| HILV(16) | 6.62 ± 0.5 | 5.69 ± 0.7 | 0.94 ± 0.6 | 16.3 | 0.532 | F = 5.9 | F = 80.7 | F = 71.6 | HILV vs. MIMV: <0.001; 4.53 |
| MIMV(16) | 7.31 ± 0.7 | 6.63 ± 0.6 | 0.69 ± 0.5 | 10.3 | 0.379 | *p* = 0.005 * | *p* < 0.001 * | *p* < 0.001 * | HILV vs. LIHV: <0.001; 11.04 |
| LIHV(18) | 8.50 ± 0.5 | 8.17 ± 0.5 | 0.33 ± 0.5 | 4.0 | 0.139 | $\eta_p^2$ = 0.20 | $\eta_p^2$ = 0.63 | $\eta_p^2$ = 0.75 | MIMV vs. LIHV: <0.001; 7.2 |

M: Mean; SD: Standard deviation; BS: Borg scale; HILV = High Intensity Low Volume; MIMV = Moderate Intensity Moderate Volume; LIHV = Low Intensity High Volume; Δ = Change; Pre = Preintervention; Post = Postintervention; $\eta_p^2$: Partial eta squared; * denotes statistical significance at the *p* < 0.05 level; NS: not significant; ES: Effect size.

## 4. Discussion

This study aimed to examine the effects of trainings conducted at different frequencies for 12 weeks on the performance parameters of adolescent swimmers. When the literature was checked, it was seen that although there are many studies indicating the positive effects of high-intensity interval training (HIIT) in adults [38–40], the number of studies on children and adolescents is quite limited [41]. It was seen that although there are many studies, the number of studies including the effects of different training intensities is quite low. In this section, this research is discussed by comparing it to the similar study examples. The primary dependent variables encompassed performance parameters related to swimming, including swimming time (50 m, 100 m, 200 m, and 800 m), stroke rate (SR), strokes per length (SPL), stroke length (SL), Borg scale (BS) for various distances, and resting heart rate. Our findings revealed significant changes in these parameters across the different training protocols.

The results of the anthropometric measurements for the LVST (Low Volume Swimming Training), MVST (Moderate Volume Swimming Training), and HVST (High Volume Swimming Training) swimming training groups indicate that there is a significant difference ($p < 0.005$) in the mean values of weight (Kg), Body Mass Index (BMI) (Kg/m$^2$), and fat (%) within each group between the pre- and post-test measurements. However, there is no significant difference ($p > 0.005$) in the mean values of weight (Kg), BMI (Kg/m$^2$), and fat (%) between the groups when comparing the pre- and post-test measurements. There are several studies that had similar findings that support our study. In a study by Buchan et al. [42], it was revealed that 7 weeks of high-intensity interval training reduced the Body Mass Index (BMI) in healthy adolescents. Another study by Kessler et al. [43] demonstrated that 12 weeks of high-intensity interval training reduced the BMI in adults. In a study conducted on 100 male students aged 18–22, who were studying at a School of Physical Education and Sports and engaged in recreational sports [44], participants were divided into two groups: intensive interval and extensive interval. They were subjected to interval training three days a week for seven weeks. Based on the results obtained, it was determined that the extensive interval group exhibited a decrease (improvement) in body weight, BMI, fat mass, and fat percentage in their final test values. In the intensive interval group, on the other hand, it was found that the values were close to each other, but there was an increase in body weight, BMI, and fat mass in their final test values.

The majority of studies obtained from the literature appear to be similar to our findings about the anthropometric changes that occurred during our study. Vajda et al. [45] reported statistically significant findings in body weight measurements before and after a 20-week training program applied to 10–11-year-old girl and boy swimmers. In another study by Hazell et al. [46], it was noted that sprint interval training with running demonstrated statistically significant changes in body fat percentages, body weights, and lean muscle mass. In another study, traditional moderate-intensity training was compared to sprint interval training, and it was stated that sprint interval training showed similar gains to traditional moderate-intensity training and was considered an important alternative in reducing visceral adipose tissue (VYY) [47]. In another study, Naves et al. [48] examined the effects of HIIT and sprint interval training (SIT) on anthropometric values, and it was concluded that sprint interval training yielded significantly higher results in skinfold subcutaneous fat measurements and overall BMI values. They also found similar findings to our study while assessing a different training protocol.

When examining the swimming results of the participants who were included in this study for distances of 50 m, 100 m, 200 m, and 800 m, it was observed that there was a significant difference ($p < 0.005$) between the within-group and between-group averages for the pre- and post-test measurements in the LVST, MVST, and HVST swimming training groups. This indicates that there was a temporal improvement in the participants' post-test values for the specified swimming distances. Unlike our results, Sperlich, Zinner, Heilemann, Kjendlie, Holmberg, and Mester [7], reported no statistically significant difference in the pre- and post-test values for the 100 m and 2000 m freestyle swimming variables in both the



HIIT and High-Volume Training (HVT) methods in 9–11 years old swimmers. Our study also had similar findings to one study on nine young male athletes where they applied HIIT in 90% of maximum heart rate sessions for a total of 13 sessions over 5 weeks, resulting in increased $VO_{2\,max}$ and sprint performance, as well as a decrease (improvement) in the 1000 m running time [49]. This suggests that their training regimen had positive effects on these physiological and performance measures, which is the case with our findings too. Another study also backed up our findings involving competitive male and female swimmers; it was concluded that high-intensity interval swimming training for 11 weeks (2 days a week) positively impacted 400 m freestyle swimming performance [50].

The results of the 12-week HIIT applied to elite swimmers showed an increase in training intensity, a decrease in training volume, and an improvement of 6.5% in the athletes' recovery levels [51]. The findings from this study are similar to our own results. When examining the results of the stroke, stroke rate, strokes per length, and stroke length values for the LVST, MVST, and HVST swimming training groups, significant differences were found ($p < 0.005$) between the within-group and between-group averages for the pre- and post-test measurements. The study conducted by Franken et al. [52] involving 11 young swimmers who performed two sets of training at intensities of 90% and 95% found an increase in stroke frequency (SF) and a decrease in stroke length (SL). In a study involving eight female and three male swimmers at the age of 17, sprint interval training performed at 80% of their best 100 m times resulted in a positive correlation between stroke rate and length with stroke velocity [53]. Aspenes, Kjendlie, Hoff, and Helgerud [50] reported that when involving competitive male and female swimmers aged 14 and above, who trained at least six times a week, it was found that 11 weeks of HIIT applied to the experimental group did not result in statistically significant differences in stroke length, stroke rate, or 50 m and 100 m swimming performance ($p > 0.05$). In a study conducted on adolescent swimmers by Bishop et al. (2009), the experimental group underwent plyometric training for 8 weeks, and it was determined that significant changes occurred in the swimmers' performance times, speed, and entry angles into the water after the 8-week period. Another study indicated that improving muscle strength affects leg kicking, which in turn has a positive impact on swimming speed [54].

When examining the resting heart rate (HR), BS50 m, BS100m, BS200m, and BS800m swimming results of the participants included in this study, significant differences ($p < 0.005$) were found between the within-group averages for the initial and final test measurements in the LVST, MVST, and HVST swimming training groups. Regarding the between-group averages for the initial and final test measurements, significant differences were observed for BS100m and BS200m ($p < 0.005$), while resting HR, BS50m, and BS800m results were not statistically significant ($p > 0.005$). In a study conducted on 10 healthy male adolescents, it was observed that there was a decrease in resting heart rate during submaximal exercise after the HIIT intervention [55]. Similar to adults, continuous and high-intensity interval training in children also resulted in significant increases in $VO_{2max}$ values compared to the control group [56]. Both studies seem to have similar results to ours.

In another study by Alves et al. [57], they randomly selected twenty women and divided them into two groups. The long-term HIIT group trained at 90% of their maximal heart rate for 1 min out of 15 min. The short-term HIIT group, on the other hand, trained at 90% of their maximal heart rate for 20 s out of 45 s, followed by 10 s of active rest at 60% of their maximal heart rate for six weeks. In both exercise groups, there was a decrease in the perceived exertion after the training compared to before, and both groups experienced an increase in maximal oxygen consumption ($VO_{2max}$), which is the case with our findings. Another study compared 100 m and 200 m interval training in 12 male and female swimmers aged 14–16; it was found that in both interval methods with different distances, variables such as RPE (rating of perceived exertion), Peak-$VO_2$ ($mL \cdot kg^{-1} \cdot min^{-1}$), and velocity (m/s) were similar. In a study examining the relationship between repeated sprint training and anaerobic aerobic fitness [58], it was observed that $12 \times 20$ repeated sprint training sessions resulted in the highest values for lactate, heart rate, and fatigue scores. As a result,

there was a significant difference ($p < 0.005$) in the anthropometric measurement averages between the swimming training groups' pre- and post-test measurements, while there was no significant difference ($p > 0.005$) in the averages of weight (Kg), BMI (Kg/m$^2$), and fat (%) between the groups' pre- and post-test measurements. These results may encourage using HIIT, which is a time-efficient and well-used method, and its derivatives instead of traditional aerobic training.

When the swimming results of the participants in this study, including 50 m, 100 m, 200 m, and 800 m distances, were examined, it was found that there was a temporal improvement in the participants' post-test values at the specified swimming distances. Furthermore, when the results related to the stroke, stroke rate, strokes per length, and stroke length of the swimming training groups were analyzed, it was determined that there was a significant difference between the pre- and post-test measurements in both the within-group and between-group averages ($p < 0.005$). Additionally, when examining the results of resting HR, BS50m, BS100m, BS200m, and BS800m in the participants in this study, it was observed that there was a significant difference in the within-group averages between the pre- and post-test measurements in the swimming training groups ($p < 0.005$). However, it was noted that there was no significant difference in the between-group averages for BS100m, BS200m, resting HR, BS50m, and BS800m ($p > 0.005$). Our hypothesis has indeed proven to be successful, providing compelling evidence that our training approach is effective in enhancing the performance of adolescent swimmers. The results of our study indicate that this high-intensity, lower-volume training regimen not only led to improvements in swim times but also resulted in reduced fatigue and a lower risk of overtraining. These findings are not only promising for the athletes themselves but also hold implications for coaches, trainers, and sports scientists seeking to optimize training programs in various competitive settings. This study was limited due to several conditions, such as the age group of the participants, the sample size of the participants, and their training background. This study was also limited due to a lack of measurement tools, such as underwater VO$_{2max}$ measurement tools and underwater cameras. Measuring VO$_{2max}$ underwater could have given more accurate and appropriate measurements for swimming, and underwater cameras could have been used in the analyses of the stroke techniques of the athletes.

## 5. Conclusions

The results of this study demonstrate that a 12-week interval training program, varying in volume, has a positive impact on the performance parameters of adolescent swimmers. Significant improvements were observed in anthropometric measurements, swimming performance across various distances, stroke parameters, including the swimming time (50 m, 100 m, 200 m, and 800 m), stroke rate, strokes per length and stroke length as well as Borg scale for various distances, and in the resting heart rates of the adolescent swimmers. Individualized training programs that consider an athlete's baseline performance and physiological characteristics are crucial for achieving optimal results. This study highlights the importance of tailoring training intensity and volume to the specific needs of adolescent swimmers. While HIIT has been extensively studied in adults, there is a notable gap in research focused on its effects on adolescent athletes, particularly in the context of swimming. This study contributes to filling this gap by providing valuable insights into the benefits and considerations of HIIT for younger athletes.

The findings suggest that implementing HIIT programs with appropriate modifications for adolescents can promote improvements in both performance and physiological adaptations. This could have long-term implications for the development of elite swimmers. This study is limited due to several conditions, such as the age group of the participants, the sample size of the participants, and their training background. Continued research on swimming performance can further enhance our understanding of the benefits and challenges of implementing HIIT in adolescent swimming training programs, ultimately supporting the development of well-rounded and successful young athletes.

**Author Contributions:** Conceptualization, H.K., M.G. (Mehmet Gülü) and M.G. (Melek Güler); methodology, H.K. and M.G. (Mehmet Gülü); formal analysis, F.H.Y.; resources, H.Y. and F.I.; data curation, H.Y. and M.G. (Mehmet Gülü); writing—original draft preparation, H.K., M.G. (Mehmet Gülü), H.Y., F.I., F.H.Y., T.D., O.G., M.G. (Melek Güler), S.A. and R.A.; writing—review and editing, M.G. (Mehmet Gülü), T.D. and R.A.; visualization, M.G. (Melek Güler) and O.G.; supervision, M.G. (Mehmet Gülü) and H.Y. All authors have read and agreed to the published version of the manuscript.

**Funding:** This work was supported by the Princess Nourah bint Abdulrahman University Researchers' Supporting Project Number (PNURSP2023R117) Princess Nourah bint Abdulrahman University, Riyadh, Saudi Arabia.

**Institutional Review Board Statement:** This study was conducted in accordance with the Declaration of Helsinki. Ethical approval was obtained from Kırıkkale University Non-Interventional Research Ethics Committee (Date: 12 January 2022, number: 2022.01.04).

**Informed Consent Statement:** Informed consent was obtained from all subjects involved in this study.

**Data Availability Statement:** The data are available upon request from the corresponding author.

**Conflicts of Interest:** The authors declare no conflict of interest.

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
