# Peer review of "Effects of 12 Weeks of High-, Moderate-, and Low-Volume Training on Performance Parameters in Adolescent Swimmers"

_applsci, doi:10.3390/app132011366_

Round 1
Reviewer 1 Report
INTRODUCTION
Authors are encouraged to summarise and review the state of the art in the introduction. There is a lot of information that should be summarised.
Lines 50-52: Please include a reference to support your statement.
Lines 53-57: Please include a reference to support your statement.
Lines 66: Add a connector to link. Start the paragraph correctly.
Lines 67-68: Please include a reference to support your statement.
Lines 81-82: Please include a reference to support your statement.
Lines 82-84: Please include a reference to support your statement.
Lines 92-96: Please include a reference to support your statement.
MATERIALS AND METHODS
Lines 124-125: Inclusion of a reference to the principles of the Helsinki Declaration.
Participants: Please indicate the number of sessions per week and the average distance swimmers were swimming before starting the study.
Line 184: According to the rules of the journal, please enter this reference correctly (Hagem, O'Keefe, Fickenscher, & Thiel, 2013).
Lines 201-202: According to the rules of the journal, please enter this reference correctly (Kennedy, Brown, Chengalur, & Nelson, 1990; Garland 201 Fritzdorf, Hibbs & Kleshnev, 2009).
Lines 222-224: This information could be discarded. To be assessed by the authors.
Following environmental conditions were set in all training sessions: Pool tempera- 222 ture: 25-29 C°, pool water temperature: 24-28 C°, pool water chlorine (free): 1-1.5 ppm, 223 absolute humidity 14 g/kg, relative humidity 40-50% and pool water PH was set to 7.2- 224 7.6.
RESULTS
In the results tables, where there were significant differences, these are indicated by an asterisk.
There is an error in the size of the letters in Table 4. Correct.
DISCUSSION
Authors are encouraged to discuss their results with the studies found in the literature. This means making comparisons, identifying differences or similarities and justifying them, explaining why they are there.
It is recommended to add a subsection according to the study variables. This would make the discussion easier to read and understand.
Lines 338, 340-341: Please add some studies, according to your statement.
it is seen that although there are many studies….
Again, it is seen that the number of studies including the effects of different training intensities is quite low.
Lines 343-360: It is recommended to discuss the results of the research with the results of other studies. Say whether or not there were differences between the studies.
Lines 361-372: Authors are advised to summarise the similarities and differences of the studies and compare them with the conditions or results of the present research.
Lines 373-387: Authors are advised to summarise the similarities and differences of the studies and compare them with the conditions or results of the present research.
Lines 388-391: Please discuss your results with those in this section.
Lines 342-412: Please summarise this section. Discuss your findings.
Lines 413-457: Authors are encouraged to discuss their results with the studies found in the literature. This means making comparisons, identifying differences or similarities and justifying them, explaining why they are there.
REFERENCES
References 23 and 24 are incorrectly placed.
Author Response
Responses to the Comments
Manuscript ID: applsci-2624179
Title: Effects of 12 Weeks High, Moderate & Low-Volume Training on Performance Parameters in Adolescent Swimmers
Dear Editor,
Thank you very much for allowing us to revise the manuscript. We would like to thank the editor and all the reviewers for their valuable comments and suggestions. Based on the feedback, we have revised our manuscript. The detailed modifications to address reviewers’ comments are provided in the following.
REVIEWER 1 (COLOURED BLUE IN THE MANUSCRIPT)
INTRODUCTION
Comments 1: Authors are encouraged to summarise and review the state of the art in the introduction. There is a lot of information that should be summarised.
Answer: First of All, thank you so much for taking time reviewing our paper. We highly appreciate your detailed feedbacks. We have summarised the introduction section as much as we could, without removing any essential points.
Comments 2: Lines 50-52: Please include a reference to support your statement.
Answer: Thank you for pointing this out. We have included following references to support our statement:
Yustres, I.; Martín, R.; Fernández, L.; González-Ravé, J. Swimming championship finalist positions on success in international swimming competitions. PloS one 2017, 12, e0187462.
Comments 3: Lines 53-57: Please include a reference to support your statement.
Answer: Thank you for pointing this out. We have included following references to support our statement:
Gillen, J.B.; Gibala, M.J. Is high-intensity interval training a time-efficient exercise strategy to improve health and fitness? Applied physiology, nutrition, and metabolism 2014, 39, 409-412.
Sperlich, B.; Zinner, C.; Heilemann, I.; Kjendlie, P.-L.; Holmberg, H.-C.; Mester, J. High-intensity interval training improves VO 2peak, maximal lactate accumulation, time trial and competition performance in 9–11-year-old swimmers. European journal of applied physiology 2010, 110, 1029-1036.
Comments 4: Lines 66: Add a connector to link. Start the paragraph correctly.
Answer: Thank you for pointing this out. We have added a connector to link and started the paragraph as following:
“In light of the significant advantages offered by interval training in enhancing swimmers' performance, it is essential to recognize that swimming is a versatile and inclusive sport that can be enjoyed by suitable for adolescents, regardless of their activity level or athletic background”
Comments 5: Lines 67-68: Please include a reference to support your statement.
Answer: Thank you for pointing this out. We have included following references to support our statement:
Bishop, D.C.; Smith, R.J.; Smith, M.F.; Rigby, H.E. Effect of plyometric training on swimming block start performance in adolescents. The Journal of Strength & Conditioning Research 2009, 23, 2137-2143.
Comments 5: Lines 81-82: Please include a reference to support your statement.
Answer: Thank you for pointing this out. We have included following references to support our statement:
Sanderson, M.; McKinlay, B.J.; Theocharidis, A.; Kouvelioti, R.; Falk, B.; Klentrou, P. Changes in inflammatory cytokines and irisin in response to high intensity swimming in adolescent versus adult male swimmers. Sports 2020, 8, 157.
Comments 6: Lines 82-84: Please include a reference to support your statement.
Answer: Thank you for pointing this out. We have included following references to support our statement:
Chatard, J.-C.; Mujika, I. Training load and performance in swimming. Biomechanics and medicine in swimming VIII 1999, 429-434.
Collette, R.; Kellmann, M.; Ferrauti, A.; Meyer, T.; Pfeiffer, M. Relation between training load and recovery-stress state in high-performance swimming. Frontiers in physiology 2018, 9, 845.
Comments 7: Lines 92-96: Please include a reference to support your statement.
Answer: Thank you for pointing this out. We have included following references to support our statement:
Hibberd, E.E.; Laudner, K.G.; Kucera, K.L.; Berkoff, D.J.; Yu, B.; Myers, J.B. Effect of swim training on the physical character-istics of competitive adolescent swimmers. The American Journal of Sports Medicine 2016, 44, 2813-2819.
MATERIALS AND METHODS
Comments 8: Lines 124-125: Inclusion of a reference to the principles of the Helsinki Declaration.
Answer: Thank you for pointing this out. We have included following references to support our statement:
Association, W.M. World Medical Association Declaration of Helsinki: ethical principles for medical research involving human subjects. Jama 2013, 310, 2191-2194.
Comments 9: Participants: Please indicate the number of sessions per week and the average distance swimmers were swimming before starting the study.
Answer: Thank you for pointing this out. We have added following paragraph to indicate the number of sessions per week and the average distance swimmers were swimming before starting the study:
“Before participating in our study, the participants were doing an average of 4000 meters of swimming training per week.”
Comments 10: Line 184: According to the rules of the journal, please enter this reference correctly (Hagem, O'Keefe, Fickenscher, & Thiel, 2013).
Answer: Thank you for pointing this out. We have corrected our references.
Comments 11: Lines 201-202: According to the rules of the journal, please enter this reference correctly (Kennedy, Brown, Chengalur, & Nelson, 1990; Garland 201 Fritzdorf, Hibbs & Kleshnev, 2009).
Answer: Thank you for pointing this out. We have corrected our references.
Comments 12: Lines 222-224: This information could be discarded. To be assessed by the authors.
Answer: Thank you for pointing this out. We have removed the related part.
RESULTS
Comments 13: In the results tables, where there were significant differences, these are indicated by an asterisk.
Answer: Thank you for pointing this out. We have indicated the significant differences using asterisk (*). We also corrected the sub-table information.
Comments 14: There is an error in the size of the letters in Table 4. Correct.
Answer: Thank you for pointing this out. We have corrected the size of the letters in Table 4.
DISCUSSION
Comments 15: Authors are encouraged to discuss their results with the studies found in the literature. This means making comparisons, identifying differences or similarities and justifying them, explaining why they are there.
Answer: Thank you for your contributions to our paper and we highly appreciate your detailed feedbacks. We have discussed and compared our findings with literature. We tended make clear explanations about the similarities and differences of our paper with other studies.
Comments 16: It is recommended to add a subsection according to the study variables. This would make the discussion easier to read and understand.
Answer: Thank you for pointing this out. We have added the following subsection to discussion:
“The primary dependent variables encompassed performance parameters related to swimming, including Swimming Time (50m, 100m, 200m, and 800m), Stroke Rate (SR), Strokes Per Length (SPL), Stroke Length (SL), Borg Scale (BS) for various distances, and Resting Heart Rate. Our findings revealed significant changes in these parameters across the different training protocols.”
Comments 17: Lines 338, 340-341: Please add some studies, according to your statement.
Answer: Thank you for pointing this out. We have included following references to support our statement:
Khammassi, M.; Ouerghi, N.; Said, M.; Feki, M.; Khammassi, Y.; Pereira, B.; Thivel, D.; Bouassida, A. Continuous moder-ate-intensity but not high-intensity interval training improves immune function biomarkers in healthy young men. The Journal of Strength & Conditioning Research 2020, 34, 249-256.
Mekari, S.; Neyedli, H.F.; Fraser, S.; O’Brien, M.W.; Martins, R.; Evans, K.; Earle, M.; Aucoin, R.; Chiekwe, J.; Hollohan, Q. High-intensity interval training improves cognitive flexibility in older adults. Brain sciences 2020, 10, 796.
Keating, S.E.; Machan, E.A.; O'Connor, H.T.; Gerofi, J.A.; Sainsbury, A.; Caterson, I.D.; Johnson, N.A. Continuous exercise but not high intensity interval training improves fat distribution in overweight adults. Journal of obesity 2014, 2014.
Comments 18: Lines 343-360: It is recommended to discuss the results of the research with the results of other studies. Say whether or not there were differences between the studies.
Answer: Thank you for contributions. We have specifically pointed out the similarities or differences of our results, compared to previous studies.
Comments 19: Lines 361-372: Authors are advised to summarise the similarities and differences of the studies and compare them with the conditions or results of the present research.
Answer: Thank you for pointing this out. We highlighted these comparisons with following statement:
“The majority of studies obtained from the literature appear to be similar with our find-ings, about the anthropometric changes occurred during the study…”
Comments 20: Lines 373-387: Authors are advised to summarise the similarities and differences of the studies and compare them with the conditions or results of the present research.
Answer: Thank you for your contributions to our paper. We have summarised the related section and focused on the focal points of studies to compare with our paper.
Comments 21: Lines 388-391: Please discuss your results with those in this section.
Answer: Thank you for your contributions to our paper. We have discussed our results accordingly.
Comments 22: Lines 342-412: Please summarise this section. Discuss your findings.
Answer: Thank you for your contributions to our paper. We have summarised the section and discussed our findings.
Comments 22: Lines 413-457: Authors are encouraged to discuss their results with the studies found in the literature. This means making comparisons, identifying differences or similarities and justifying them, explaining why they are there.
Answer: Thank you for pointing your contributions to our paper. We have revised this section and discussed our findings with comparisons to literature.
REFERENCES
Comments 22: References 23 and 24 are incorrectly placed.
Answer: Thank you for pointing this out. We have corrected the placement of references you pointed out.

Reviewer 2 Report
Article
Effects of 12 Weeks High, moderate& low-Volume Training on Performance Parameters in Adolescent Swimmers
Revision
The article provides an important approach to different training volumes (low, moderate and high training volumes) on performance parameters in adolescent swimmers. I suggest some manuscript considerations.
Abstract:
I would like you to mention the number of participants.
The sentence (Low Volume Swimming Training, Moderate Volume Swimming Training, and High-Volume Swimming Training) could be summarised to avoid repetition of words.
The acronym "BMI" is used without prior definition of its meaning (Body mass index).
The abstract states that significant differences were found, but does not mention what level of significance is being adopted.
The variables used in the results could have been mentioned more briefly, for example: stroke rate and length.
Introduction
There really are few studies dealing with the theme proposed in this manuscript. I understood the authors' logic in the introduction. My question is whether they could write the hypothesis of the study at the end of the introduction.
I suggest that the authors write in the discussion the result for the hypothesis assumed.
Materials and Methods
Line 136 – The acronym "BMI" is used without prior definition of its meaning (Body mass index).
Line 137 - where it reads "m2", write "2" in superscript.
Line 142 - where it reads posttest, write "post-test".
Use the same pattern of terms throughout the manuscript. Check throughout the manuscript.
Discussion
Lines 350 and 351 - Was this the first time you wrote the name of the acronym? Use the full acronym name the first time you write it in the text (Body Mass Index BMI). After that sentence, just write BMI.
I missed the limitations of this manuscript. Do the authors want to declare any limitations in this study? I congratulate them on their work, including the sample size assumed.
Conclusions
The conclusion is very good, but I realise that it is a little generic. I suggest that you incorporate the variables analysed into the text of the conclusion. This suggestion should be applied because the conclusion of this manuscript addresses specific issues according to the objective of the manuscript.
Author Response
Responses to the Comments
Manuscript ID: applsci-2624179
Title: Effects of 12 Weeks High, Moderate & Low-Volume Training on Performance Parameters in Adolescent Swimmers
Dear Editor,
Thank you very much for allowing us to revise the manuscript. We would like to thank the editor and all the reviewers for their valuable comments and suggestions. Based on the feedback, we have revised our manuscript. The detailed modifications to address reviewers’ comments are provided in the following.
REVIEWER 2 (COLOURED BLUE IN THE YELLOW)
ABSTRACT
Comments 1: I would like you to mention the number of participants.
Answer: First of All, thank you so much for taking time reviewing our paper. We highly appreciate your detailed feedbacks. We have added the number of participants to abstract section of our paper.
Comments 2: The sentence (Low Volume Swimming Training, Moderate Volume Swimming Training, and High-Volume Swimming Training) could be summarised to avoid repetition of words.
Answer: Thank you for pointing this out. We have revised this part as following:
“High Intensity Low Volume (HILV), Moderate Intensity Moderate Volume (MIMV), Low Inten-sity High Volume (LIHV)”
Comments 3: The acronym "BMI" is used without prior definition of its meaning (Body mass index).
Answer: Thank you for pointing this out. We have revised this part as following:
“…Body Mass Index (BMI,)…”
Comments 4: The abstract states that significant differences were found, but does not mention what level of significance is being adopted.
Answer: Thank you for pointing this out. We have added the level of significance to stated differences.
Comments 5: The variables used in the results could have been mentioned more briefly, for example: stroke rate and length.
Answer: Thank you for pointing this out. We have mentioned all variables that we used in the results briefly.
INTRODUCTION
Comments 6: There really are few studies dealing with the theme proposed in this manuscript. I understood the authors' logic in the introduction. My question is whether they could write the hypothesis of the study at the end of the introduction.
I suggest that the authors write in the discussion the result for the hypothesis assumed.
Answer: Thank you for pointing this out. We have added our hypothesis in the end of the introduction section and discussed the results accordingly to our hypothesis. Our hypothesis is as following:
“In the context of this research, our hypothesis centered around the notion that employing a training regimen characterized by high-intensity workouts paired with a reduced volume of training sessions would yield more favorable outcomes among adolescent swimmers. We believed that this unique approach to training had the potential to offer significant ad-vantages and improvements when compared to conventional training methods commonly employed in the sport of swimming for this age group. By emphasizing high-intensity exercises while reducing the overall training load, we aimed to uncover the potential benefits and optimize performance for young athletes.”
MATERIALS AND METHODS
Comments 7: Line 136 – The acronym "BMI" is used without prior definition of its meaning (Body mass index).
Answer: Thank you for pointing this out. We have revised this part as following:
“….Body Mass Index (BMI)…”
Comments 8: Line 137 - where it reads "m2", write "2" in superscript.
Answer: Thank you for pointing this out. We have written the “2” in superscript as m2.
Comments 9: Line 142 - where it reads posttest, write "post-test".
Answer: Thank you for pointing this out. We have corrected this error and wrote as following:
“…pre-test and post-test…”
Comments 10: Use the same pattern of terms throughout the manuscript. Check throughout the manuscript.
Answer: Thank you for pointing this out. We have checked the terms we used in the study all over. We applied the same pattern of terms throughout the study.
DISCUSSION
Comments 11: Lines 350 and 351 - Was this the first time you wrote the name of the acronym? Use the full acronym name the first time you write it in the text (Body Mass Index BMI). After that sentence, just write BMI.
Answer: Thank you for pointing this out. We have corrected this error in discussion section and applied it to all manuscript.
Comments 12: I missed the limitations of this manuscript. Do the authors want to declare any limitations in this study? I congratulate them on their work, including the sample size assumed.
Answer: Thank you for your contributions and precious comments. We have added our limitations to discussion section as following:
“This study was limited due to several conditions such as the age group of participants, sample size of participants and their training background. This study is also limited due to lack of measurement tools such as underwater VO2max measurement tools and under-water cameras. Measuring VO2max underwater could have given more accurate and appropriate measurements for swimming and underwater cameras could have use in the analyses of stroke techniques of athletes.”
CONCLUSIONS
Comments 12: The conclusion is very good, but I realise that it is a little generic. I suggest that you incorporate the variables analysed into the text of the conclusion. This suggestion should be applied because the conclusion of this manuscript addresses specific issues according to the objective of the manuscript.
Answer: Thank you for your great evaluations. We have specifically addressed our variables in this section. We tried to give more direct conclusions according to our results.

Round 2
Reviewer 1 Report
Dear Authors,
Thank you for making the changes to your manuscript.
Kind regards
Author Response
Thank you very much for allowing us to revise the manuscript. We would like to thank the editor and all the reviewers for their valuable comments and suggestions.